# Revisiting the Green Growth Effect of Foreign Direct Investment from the Perspective of Environmental Regulation: Evidence from China

**DOI:** 10.3390/ijerph20032655

**Published:** 2023-02-01

**Authors:** Deyun Xiao, Luyao Gao, Lijia Xu, Zongjun Wang, Wu Wei

**Affiliations:** 1School of Economics, Wuhan University of Technology, Wuhan 430070, China; 2School of Management, Huazhong University of Science and Technology, Wuhan 430074, China; 3School of Economics and Management, Wuhan University, Wuhan 430072, China

**Keywords:** green economic growth, foreign direct investment, independent innovation, imitation innovation, environmental regulation

## Abstract

The inflow of foreign direct investment (FDI) has both advanced China’s economic development process and influenced the ecological quality of China’s regions. Under the deepening of economic globalization and the continuous deterioration in environmental quality, the correlation mechanism between foreign direct investment, environmental regulation, and economic growth is becoming increasingly complex. Therefore, based on the slacks-based measure (SBM) model and the Global Malmquist-Luenberger (GML) index, this study measured the level of green economic growth using data from 30 provinces and cities from 2004–2019 and constructed a panel fixed-effect regression model to study the effect of foreign direct investment on green economic growth in China. The study found that foreign direct investment significantly promoted green economic growth in China, foreign direct investment promoted green economic growth through independent innovation and inhibited green economic growth through imitation innovation, and environmental regulation moderated the impact of foreign direct investment on green economic growth. This paper incorporated foreign direct investment, heterogeneous technological innovation, green economic growth, and environmental regulation into the research framework, and thereby further enriched and improved the research on the theoretical mechanism of green economic growth. The research conclusion clarified the influence mechanism of foreign capital on the quality of China’s economic development, which was conducive to the formulation of more reasonable policies for attracting investments and to the promotion of the formation of a positive interaction mechanism between environmental regulation and foreign direct investment, which is of great practical significance for China’s economy to achieve sustainable development.

## 1. Introduction

In the past 42 years of reform and opening up, China’s economy has developed rapidly, but in recent years, China’s economic growth has encountered two difficulties: economic growth is slowing down, and environmental pollution is serious. The previous economic development model has resulted in the rapid consumption of resources and serious pollution of the ecological environment and can no longer strongly support China’s sustained economic development. To break through these two dilemmas, China has firmly committed to a sustainable development strategy, thereby making it clear that the current economic development model should be carried out without compromising the development needs of future generations, and that economic growth should be achieved while reducing resource consumption and protecting the ecological environment [1]. The inflow of foreign capital will certainly have an important impact on the green transformation of China’s economy [2], and foreign direct investment (FDI) has brought huge amounts of capital and advanced technology to China, thereby promoting rapid technological progress and economic growth. Meanwhile, given the diversity of technological innovation models, the impact of FDI on China’s green economic growth under different technological paths is uncertain [3]. As the construction of ecological civilization has advanced, the regional governments in China have introduced environmental regulation policies that impose certain constraints and provide guidance on the scale and quality of FDI inflows and the production management mode of domestic enterprises [4]. In the context of globalization and China’s green economic transformation, it is of great theoretical and practical significance to explore the impact of FDI on China’s green economic growth under the influence of environmental regulation and to further study the role mechanisms of independent innovation and imitation innovation therein.

In recent years, a large number of experts and scholars have begun to pay attention to the impact of FDI on the economic development of the host country, but the existing research has not yet reached a consistent conclusion on the impact of FDI on green economic growth. The Organization for Economic Co-operation and Development defines green economic growth as the promotion of economic growth and development while ensuring that natural assets continue to provide a variety of resource and environmental services for human well-being. Some scholars believe that FDI has a “pollution halo” effect; that is, it can have a positive impact on the quality of economic development of the host country through economic agglomeration and structural optimization. FDI has brought advanced green technology and production equipment from the home country through technology licensing, technology spillover, and other methods of driving the host country’s technological progress, resource conservation, and green production [5]. Some scholars also believe that FDI from developed countries mainly flows into resource-intensive and pollution-intensive industries in the host country, thereby destroying the environmental quality of the host country; that is, there is a “pollution haven” phenomenon [6]. Based on the existing literature, this study incorporated foreign direct investment, heterogeneous technological innovation, green economic growth, and environmental regulation into the research framework with an aim to address two major issues: how FDI affects green economic growth under different technological innovation paths and how environmental regulation regulates the mechanism of FDI’s effect on green economic growth. This study comprehensively analyzed the mechanism of the impact of FDI on green economic growth under environmental regulation and introduced heterogeneous technological innovation to analyze its mediating effect in FDI and green economic growth, which will enrich the previous research in the field of FDI and green economic growth.

The rapid consumption of energy and the continuous deterioration in the environment are the two major problems in the process of the strategic restructuring of China’s economy. In the context of the current economic environment, this study can help to clarify more clearly the impact of foreign investment on the quality of China’s economic development, which will be conducive to the formulation of more reasonable policies for attracting investments and promoting the formation of a positive interaction mechanism between environmental regulation and FDI and also is of great practical significance in solving the difficult problems encountered in China’s economic transformation.

Based on foreign direct investment theory, innovation theory, and economic growth theory, this study examined the impact of FDI and heterogeneous technology innovation model on green economic growth under environmental regulation. The study used a slacks-based measure (SBM) directional distance function and the Global Malmquist-Luenberger (GML) index to measure China’s provincial green total factor productivity as a measure of regional green economic growth as well as the panel data of 30 provinces and municipalities in China from 2004 to 2019 to construct a panel fixed-effect regression model to test the impact mechanism and transmission path of FDI on regional green development. Compared with the existing literature, this paper attempted to expand on the following three aspects:

(1)Measurement of the green total factor productivity (GTFP): this paper used the SBM directional distance function and GML index to measure the green total factor productivity using the input (labor, capital, and energy), expected output (GDP), and unexpected output (wastewater, industrial solid waste, carbon dioxide emissions, etc.) to effectively measure the level of regional green economic development.(2)Research on the regulation effect of environmental regulation: in the context of continuous economic globalization and continuous deterioration in environmental quality, the correlation mechanism between FDI and the host country’s environmental regulation becomes increasingly complex, so this paper examined the impact of the interaction between environmental regulation and FDI on green economic growth.(3)Research on the mediating effect of technological innovation: previous scholars have studied the impact of FDI on technological innovation or the impact of FDI on green economic growth but have failed to integrate the three into a systematic analysis framework. This study examined the transmission mechanism of FDI on green economic growth through heterogeneous technological innovation and built a research framework regarding how FDI influences different modes of technological innovation and thus differently affects green economic growth.

## 2. Literature Review

Previous scholars have conducted extensive research based on FDI, technological innovation, and green economic development.

### 2.1. FDI and Green Economic Growth

The Organization for Economic Co-operation and Development defines green economic growth as the promotion of economic growth and development while ensuring that natural assets continue to provide a variety of resource and environmental services for human well-being. Green economic growth emphasizes maximizing economic growth with minimal resource consumption and environmental costs.

The positive promotion effects of FDI on China’s upgrading of industrial structures, technological progress, employment, and economic growth have been verified by many domestic scholars [7,8,9]. FDI can bring advanced production technology to host countries (especially developing countries); however, environmental problems are also becoming increasingly prominent. Regarding the relationship between FDI and green economic development, the domestic and foreign research conclusions are not uniform. Imran et al. [10] argued that FDI is a source of environmental degradation that increases domestic carbon emissions, which would confirm the “pollution paradise” hypothesis that FDI flows into resource-intensive, pollution-intensive industries in developing countries and destroys the environmental quality of the host country. Some scholars have questioned the “pollution paradise” hypothesis by arguing that FDI inflows stimulate regional economic growth and reduce the intensity of air pollution and that foreign investment is a potential pillar to achieve the goals of green growth strategies and enhance the development of a country’s green economy [11].

### 2.2. FDI and Technological Innovation

Using firm-level data from Taiwan, Chuang and Lin [12] confirmed that FDI and R&D had a positive impact on productivity. Zhang and Jin [13] argued that FDI effectively promotes patent development in China under open economic conditions. Some scholars have argued that FDI does not have a significant relationship with technological progress in the host country and even inhibits the innovation capacity of domestic firms [14]. In particular, Wang and Zhao [15] pointed out that there is a difference in the role of FDI in different modes of technological innovation. Yang and Liu [16] argued that if the level of technology spillover is low, firms with strong domestic innovation capabilities will choose to innovate on their own; while if the level of FDI technology spillover is high, then domestic firms—regardless of their capabilities—will prefer to achieve technological progress through the low-cost and fast-imitation innovation route. Xing and Zhang [17] showed that if the government can effectively intervene in the form of technology transfer from foreign capital, then domestic firms will consider more factors to improve their imitative innovation capability and achieve technological catch-up at a lower cost.

### 2.3. Technological Innovation and Green Economic Growth

Most scholars agree that technological progress can promote green economic growth [18,19], but the impact of different technological innovation models on economic development should be discussed separately. Some scholars have argued that imitation innovation is more conducive to green economic development than independent innovation. Based on the high sunk cost and high risk of failure of independent innovation [20], He and Fan [3] pointed out that independent innovation is more concerned with capacity enhancement and ignores the environmental benefits of innovation; while imitation innovation, as a type of the following innovation, will intentionally or unintentionally follow the international advanced technology in the direction of clean and environmental protection, which is conducive to high-quality economic development. Some scholars hold the opposite view that if we continue to rely only on imitation innovation, the economic growth rate will eventually tend to be lower, and the improvement in independent innovation capacity will help to transform the economic growth mode [21].

At present, many scholars have conducted extensive and in-depth research on the relationship between FDI and green economic growth, but the specific paths of FDI have not been explored in depth. The existing studies only focused on the impact of FDI on technological innovation or green economic growth, failed to integrate the three into a systematic analysis framework, and did not analyze technological innovation in a more detailed manner; in addition, there are few studies on the regulatory role of environmental regulation in the overall framework [19,22]. This study examined both the direct moderating effect of environmental regulation on FDI and green economic growth and the mediating moderating effect of environmental regulation through mediating variables. This study also examined the intermediary transmission mechanism of FDI to green economic growth through heterogeneous technological innovation and built a research framework regarding how FDI affects different modes of technological innovation and then affects green economic growth differently. At the same time, the study examined the effect of environmental regulation on FDI and green economic growth.

## 3. Theoretical Analysis and Hypothesis Proposals

### 3.1. The Influence of FDI on Green Economic Growth

FDI mainly contributes to the level of green economic growth through the capital effect, technology spillover effect, technology transfer effect, and environmental effect [23,24].

The capital effect of FDI can be divided into the crowding-in effect and the crowding-out effect. The crowding-in effect is mainly reflected in the bridging of the savings gap, foreign exchange gap, and tax gap [25]. FDI can realize capital crowding-in through greenfield construction, cross-border M&A, profit reinvestment, and additional investments, which can help improve the asset pattern of China and form high-quality capital. The crowding-out effect is mainly reflected in the low level of technology and lack of core competitiveness of domestic enterprises compared with foreign enterprises, which make it difficult to compete with foreign enterprises and result in gradually being squeezed out of the market [26].

The technology spillover effect of FDI mainly refers to the passive diffusion or spillover of technology from foreign enterprises to the host enterprises, which brings about the technological progress of domestic enterprises and affects economic growth [23]. In the process of multinational investment and retaining business, multinational enterprises will cause demonstration effects in domestic enterprises, which will promote market prosperity and green economic growth under the premise of healthy competition. The foreign enterprises will employ local labor in the host country and provide training to ensure normal production and operation activities in the host country, and then the mobility in the local labor market can transmit advanced ideas to domestic enterprises. Moreover, the inflow of FDI promotes the technological progress and green economic growth of the entire industrial chain.

Technology transfer is mainly in the form of transfer from the parent company to overseas subsidiaries and from the parent and subsidiaries to domestic enterprises [27]. Based on the theory of internalization advantage and considering that the intellectual property protection system of the host country may not be perfect, the investing enterprises often choose to transfer their assets and technologies to their subsidiaries in the host country via internalization to protect their intellectual property rights and ensure that their patented technologies are not imitated and learned to the greatest extent. In addition, foreign-funded enterprises will also take the initiative to transfer advanced technology to domestic enterprises through technology transfer and licensing to promote the upgrading of local industrial structure and achieve benign economic agglomeration and technological progress.

High-quality foreign capital enters the local green and clean high-tech industries and imports the green technology and green international environmental standards of the investing country into the host market, which then increases the proportion of local high-value-added and high-tech industries and optimizes the ecological environment of the host country. The awareness of environmental protection in developed countries is relatively high, and foreign enterprises that enter China introduce advanced green technologies and pollution control technologies that are transferred to domestic enterprises via technology spillover and technology transfer to reduce environmental pollution through green production in the host country with environmentally friendly technologies [11].

In summary, while considering that the foreign investment attracted by China has greatly enhanced economic strength and green technology progress, improved resource utilization efficiency, and reduced pollution emissions, we proposed the following hypothesis:

 **Hypothesis 1 (H1).** 
*FDI promotes green economic growth in China.*


### 3.2. The Mediating Effect of Technological Innovation

#### 3.2.1. The Influence Mechanism of FDI in Technological Innovation

FDI mainly affects technological innovation in host countries through the capital effect, technology spillover effect, and technology transfer effect [23,24,25,26]. FDI promotes the improvement of the domestic capital market and relieves the pressure of financing so that the host country has more abundant capital to purchase advanced production equipment and conduct more frequent R&D and innovation activities, which will improve the overall innovation capacity of a country. Foreign-funded enterprises entering the domestic market—whether to realize the protection of core intellectual property rights, transfer production technology to subsidiaries in the form of internalization, or maximize their interests—through technology licensing, technical assistance, and other forms of technology transfer to domestic traditional enterprises can help China’s domestically funded enterprises via technological innovation, eliminate backward technology, optimize resource allocation, and improve total factor productivity. With the emergence of demonstrations, personnel flow, industrial linkages, and the competition effect, domestic enterprises constantly absorb the advanced technology of foreign capital spillover; carry out corresponding learning, reference, and imitation; and realize the improvement in the levels of imitation innovation and independent innovation based on imitation.

#### 3.2.2. The Influence Mechanism of Technological Innovation in Green Economic Growth

Imitation innovation refers to the innovation behavior of enterprises in imitating the existing advanced technology in the market based on the understanding of domestic market demand [21]. Imitation innovation helps enterprises to control the risk of R&D innovation and achieve economic efficiency with the lowest cost and risk. However, imitation innovation also harms China’s green economic growth. First of all, imitation innovation results in retaining the status of technology follower: the leading enterprises have already occupied most of the market share, and the imitators always obtain fewer benefits than the enterprises with the first-mover advantage. Secondly, the environmental cost of the imitation innovation of backward technology is too high. Domestic enterprises spread high-energy-consuming and high-pollution production technologies through imitation and innovation, which leads to the further destruction of the domestic environment. Finally, to continuously guarantee their technological monopoly advantage in the market, enterprises with advanced technology will try their best to block the core technology and set up trade barriers to reduce the technology spillover, thereby making it difficult for imitating enterprises to access and master the core technology in time.

Independent innovation is the original innovation carried out by enterprises based on their initiative [3]. Independent innovation requires a certain degree of knowledge accumulation, and the R&D process requires continuous high-intensity investment in capital and human resources and long-term exploration and development, so the risk of independent innovation is greater than that of imitation innovation. Although the initial investment for the cultivation of independent innovation capability is large, the latter benefit is considerable. Independent innovation can help reduce China’s technological dependence on foreign capital and break through the high-end technology blockade of developed countries. In addition, the independent innovation behavior of enterprises is based on their development, and the technologies they develop are the most suitable for their own needs, which helps to optimize the efficiency of resource allocation in the production process.

Based on the above analysis, it can be seen that FDI will directly affect the growth of the green economy and will also indirectly affect the growth of the green economy by acting on independent innovation and imitation innovation; that is, the partial impact of FDI on the growth of the green economy can be achieved through the intermediary transmission path of independent innovation and imitation innovation. Due to the different effects of independent innovation and imitation innovation on green economic growth, the following hypothesis was proposed:

 **Hypothesis 2 (H2).** 
*FDI promotes green economic growth through independent innovation and inhibits green economic growth through imitation innovation.*


### 3.3. Moderating Effects of Environmental Regulation

The “compliance cost effect” suggests that when the government implements environmental regulation policies, enterprises’ production and operation costs increase, which will crowd out the investment in R&D and innovation that belongs to enterprises, thus affecting their innovation capacity and leading to their technological backwardness and a loss in competitiveness [4]. The “innovation compensation effect” shows that environmental regulation can help enterprises to recognize the problems of resource waste and technological backwardness in their production activities, accelerate resource allocation optimization and technological improvement, and enhance their R&D and innovation capabilities in green technologies. Environmental regulation will have a positive screening effect on FDI. A strict environmental regulation level will eliminate low-quality foreign capital, give priority to the introduction of clean foreign capital conducive to technological upgrading and environmental protection, and squeeze out FDI representing backward technology that attempts to flow into China’s pollution-intensive industries. Specifically, strict environmental regulation will promote the positive spillover of FDI capital, technology transfer, the technology spillover effect, and the environmental effect; introduce advanced environmental protection concepts and green technologies; expand the scale of domestic economic activities; form economies of scale; promote the green upgrading of industrial structure; promote clean production by domestic enterprises; and promote the level of technological innovation in China. It has a catalytic effect on green economic growth.

According to the above analysis, the corresponding hypotheses were proposed:

 **Hypothesis 3 (H3).** 
*Environmental regulation has a moderating effect on the relationship between FDI and green economic growth: the greater the degree of environmental regulation, the greater the effect of FDI on green economic growth.*


 **Hypothesis 4 (H4).** 
*The stronger the environmental regulation is, the more significant the contribution of FDI to imitation innovation and the more significant the impact on the performance of green economic growth will be.*


 **Hypothesis 5 (H5).** 
*The greater the intensity of environmental regulation, the more significant the role of FDI in promoting independent innovation and the more obvious the impact on the performance of green economic growth will be.*


In summary, environmental regulation affects the impact of FDI on China’s green economic growth, and a portion of the intended effects of the regulation is achieved through the intermediary of technological innovation. The specific mechanism is shown in Figure 1:

## 4. Methodology

### 4.1. Model Building

This study used a two-way fixed-effect model for regression that considered both the individual fixed effects and time fixed effects. Since there are individual differences among provinces in terms of their industrial structures, natural environments, and development trajectories, the individual fixed-effect model solved the problem of missing variables that varied with individuals. At the same time, since each year presents different development characteristics, the time fixed effect could solve the problem of missing variables that changed over time. The regression model constructed in this paper was as follows.

Model (1) was constructed to test Hypothesis H1 and verify the impact of FDI on green economic growth.
(1)$$GTFP_{it}=\beta_{0}+\beta_{1}FDI_{it}+\sum_{j=1}^{n}\beta_{j}control_{it}^{j}+\mu_{t}+\lambda_{i}+\varepsilon_{it}$$
where $GTFP_{it}$ represents the level of green economic growth in region *i* in period *t*, $FDI_{it}$ denotes foreign direct investment in region *i* in period *t*, and $\sum control_{it}^{j}$ denotes the control variables (including market size (*SCALE*), infrastructure security (*INFRA*), government support (*GOVER*), openness to the outside world (OPEN), education level (*EDU*), degree of nationalization (*SOE*), and level of economic development (*PGDP*)). The following model control variables are the same: $\mu_{t}$ and $\lambda_{i}$ represent the fixed effects of years and provinces, respectively; and $\varepsilon_{it}$ represents the error term.

Models (2) and (3) were constructed to test Hypothesis H2 in conjunction with Model (1) to verify the mediating effect of technological innovation on FDI and green economic growth. Specifically, Model (2) was used to test the effect of FDI on independent and imitative innovation, and Model (3) was used to test the effect of independent and imitative innovation on green economic growth. *Tech_it_* represented the technological innovation in region *i* at time *t*, which was subdivided into independent innovation (*INNO_it_*) and imitation innovation (*IMMI_it_*). When *Tech_it_* represented independent innovation (*INNO_it_*), the models were used to test the mediating effect of independent innovation on FDI and green economic growth. When *Tech_it_* represented imitation innovation (*IMMI_it_*), the models were used to test the mediating effect of imitation innovation on FDI and green economic growth.
(2)$$Tech_{it}=x_{0}+x_{1}FDI_{it}+\sum_{j=1}^{n}x_{j}control_{it}^{j}+\mu_{t}+\lambda_{i}+\varepsilon_{it}$$
(3)$$GTFP_{it}=\partial_{0}+\partial_{1}FDI_{it}+\partial_{2}Tech_{it}+\sum_{j=1}^{n}\partial_{j}control_{it}^{j}+\mu_{t}+\lambda_{i}+\varepsilon_{it}$$
where *Tech_it_* refers to *INNO_it_* and *IMMI_it_*; the same applies to the below.

Models (4) and (5) introduced environmental regulation and the interaction term of FDI and environmental regulation, tested Hypothesis H3, and verified the moderating effect of environmental regulation between FDI and green economic growth (where $ER_{it}$ denotes the level of environmental regulation in period *t* in region *i*).
(4)$$GTFP_{it}=\alpha_{0}+\alpha_{1}FDI_{it}+\alpha_{2}ER_{it}+\sum_{j=1}^{n}\alpha_{j}control_{it}^{j}+\mu_{t}+\lambda_{i}+\varepsilon_{it}$$
(5)$$GTFP_{it}=\gamma_{0}+\gamma_{1}FDI_{it}+\gamma_{2}ER_{it}+\gamma_{3}FDI_{it}*ER_{it}+\sum_{j=1}^{n}\alpha_{j}control_{it}^{j}+\mu_{t}+\lambda_{i}+\varepsilon_{it}$$

Models (6) and (7) were combined with Model (5) to test Hypotheses H4 and H5 using a mediated moderation model to verify whether the moderating effect of environmental regulation was achieved through the mediating effect of technological innovation.
(6)$$Tech_{it}=\theta_{0}+\theta_{1}FDI_{it}+\theta_{2}ER_{it}+\theta_{3}ER_{it}*FDI_{it}+\sum_{j=1}^{n}\theta_{j}control_{it}^{j}+\mu_{t}+\lambda_{i}+\varepsilon_{it}$$
(7)$$GTFP_{it}=\delta_{0}+\delta_{1}FDI_{it}+\delta_{2}ER_{it}+\delta_{3}FDI_{it}*ER_{it}+\delta_{4}Tech_{it}+\sum_{j=1}^{n}\delta_{j}control_{it}^{j}+\mu_{t}+\lambda_{i}+\varepsilon_{it}$$

### 4.2. Variable Definitions

#### 4.2.1. Explained Variables

Green economic growth (GTFP): a large number of scholars use total factor productivity (TFP) as an indicator to measure economic growth. However, TFP does not take into account the problems of energy consumption and environmental pollution caused by economic development and has a certain deviation from the actual quality measurement of economic growth. For this reason, scholars have begun to consider energy input and pollution emissions in the calculation framework of TFP. Based on the traditional concept of economic growth, this paper used the data envelopment analysis (DEA) method to measure the green total factor productivity and took it as the core index to measure the level of green economic growth by considering undesirable outputs such as industrial wastewater, industrial waste gas, and industrial solid waste. Combined with data from previous studies, the paper measured the dynamic changes of green total factor productivity of 30 provinces, municipalities, and autonomous regions in China from 2000 to 2019 by considering the global directional SBM model with non-radial and non-angular non-desired outputs and the global GML productivity index to measure the green economic growth in China. Based on the research methods of Fukuyama [28], this paper constructed the SBM directional distance function as follows:(8)$$S_{V}^{G}\left(x^{k,t},y^{k,t},b^{k,t},g^{x},g^{y},g^{b}\right)=\underset{S^{x},S^{y},S^{b}}{\max}\left[\frac{1}{p}\sum_{p=1}^{P}\frac{s_{p}^{x}}{g_{p}^{x}}+\frac{1}{M+1}\left(\sum_{m=1}^{M}\frac{s_{m}^{y}}{g_{m}^{y}}+\sum_{n=1}^{N}\frac{s_{n}^{b}}{g_{n}^{b}}\right)\right]/2\;s.t.\;\sum_{t=1}^{T}\sum_{k=1}^{K}\lambda_{k}^{t}x_{kp}^{t}+s_{p}^{x}=x_{kp}^{t},\forall p;\;\sum_{t=1}^{T}\sum_{k=1}^{K}\lambda_{k}^{t}y_{km}^{t}-s_{m}^{x}=y_{km}^{t},\forall m;\;\sum_{t=1}^{T}\sum_{k=1}^{K}\lambda_{k}^{t}b_{kn}^{t}+s_{n}^{b}=b_{kn}^{t},\forall n;\;\sum_{k=1}^{K}\lambda_{k}^{t}=1,\lambda_{k}^{t}\geq 0,\forall k;\;s_{p}^{x}\geq 0,\forall p;s_{m}^{y}\geq 0,\forall m;s_{n}^{b}\geq 0,\forall n$$
where $S_{V}^{G\;}$ represents the directional distance function under the condition of variable returns to scale; $S_{C}^{G\;}$ represents the directional distance function under the condition of constant returns to scale; $\left(x^{k,t},y^{k,t},b^{k,t}\right)$ represents the input factors, expected output, and undesired output variables of province *i* in period *t*; $\left(g^{x},g^{y},g^{b}\right)$ is the direction vector, which represents the decrease in input, the increase in expected output, and the decrease in undesired output; and $\left(S^{x},S^{y},S^{b}\right)$ is the relaxation vector, which represents the excess of input, the deficiency of expected output, and the redundancy of undesired output. While referring to Oh [29], this paper constructed the GML index with the SBM model.
(9)$$GML_{t}^{t+1}=\frac{1+S_{V}^{G}\left(x^{t},y^{t},b^{t};g\right)}{1+S_{V}^{G}\left(x^{t+1},y^{t+1},b^{t+1};g\right)}$$

As the GML index reflects the growth rate of green total factor productivity, we assumed that the green total factor productivity in 2004 was 1 and then calculates the green total factor productivity from 2004 to 2019 by multiplying the GML index by the GTFP of the last year [1]. The measurement indicators of green total factor productivity are shown in Table 1.

#### 4.2.2. Explanatory Variable

Foreign direct investment (FDI): the explanatory variable used in this paper was foreign direct investment, which was measured using the ratio of the actual utilization of foreign investment to the GDP in each province. The data were derived from the *China Statistical Yearbook*.

#### 4.2.3. Moderating Variables

Environmental regulation (ER): referring to the research of Yuan [4], we selected three indicators (comprehensive utilization rate of solid waste, removal rate of sulfur dioxide, and compliance rate of wastewater discharge) and then calculated the comprehensive index of environmental regulation by entropy method. The index construction method was as follows: firstly, the three single indexes were standardized; that is, the values of each index were converted into the value range of [0, 1] via mathematical transformation. The calculation formula was as follows:(10)$$PR_{ij}^{s}=\left[PR_{ij}-min\left(PR_{j}\right)]/[max\left(PR_{j}\right)-min\left(PR_{j}\right)\right]$$
where *i* refers to the province (*i* = 1, 2, 3, …, 30); *j* refers to each single index (*j* = 1, 2, 3); *PR_ij_* is the value of *j* index in *i* province; *max*(*PR_j_*) and *min*(*PR_j_*) are the maximum and minimum values of the three indicators in all provinces, respectively; and $PR_{ij}^{s}$ is the standardized value of each indicator.

Secondly, it was necessary to calculate the weight of each index. The calculation method was as follows:(11)$$\omega_{ij}=PR_{ij}/\bar{PR_{j}}$$
where $\omega_{ij}$ refers to the weight of index *j* in province *i*, and $\bar{PR_{j}}$ is the national average level of index *j*.

Finally, the environmental regulation intensity of each province was calculated using the following formula:(12)$$ER=\frac{1}{3}\sum_{j=1}^{3}\omega_{ij}*PR_{ij}^{s}$$

#### 4.2.4. Mediating Variables

Technological innovation: in this paper, based on the research of scholars such as Gao and Di [30,31], we divided the technology sources into independent innovation (*INNO*) and imitation innovation (*IMMI*). Independent innovation was measured by the number of patents granted for inventions, which was the most innovative input factor, while imitation innovation was measured by the sum of the number of patents granted for appearance and utility patents.

#### 4.2.5. Control Variables

The control variables included market size (*SCALE*), infrastructure (*INFRA*), government support (*GOVER*), the degree of openness (*OPEN*), the level of education (*EDU*), the degree of nationalization (*SOE*), and the level of economic development (*PGDP*). Market size (*SCALE*) was measured according to the total population of each region. The infrastructure (*INFRA*) determined the convenience of regional production activities and ultimately affected GTFP, which was measured according to the ratio of the total post and telecommunications business to the GDP; government support (*GOVER*) could promote local technological progress and encourage enterprises to innovate in the form of financial subsidies and tax incentives, which was measured according to the proportion of science and technology expenditures in regional fiscal expenditures; the degree of openness (OPEN) was measured according to the ratio of total imports and exports of business units to the GDP in each region; the level of education (EDU) was measured according to the average number of years of education = (the number of people with literacy skills × 1 + the number of people with elementary school education × 6 + the number of people with junior high school education × 9 + the number of people with high school and junior college education × 12 + the number of people with college and undergraduate education × 16)/total population over 6 years old; the degree of nationalization (SOE) was measured according to the number of state-controlled industrial enterprise units/number of industrial enterprise units; and the level of economic development (PGDP) was measured according to the GDP per capita.

### 4.3. Data Source

Given the availability and completeness of some data, we selected the panel data of 30 provinces, municipalities, and autonomous regions in mainland China from 2004 to 2019 for multiple regression. All data were obtained from the National Bureau of Statistics, the *China Statistical Yearbook on Science and Technology*, the *China City Statistical Yearbook*, and the EPS data platform. Data in U.S. dollars were converted to the same unit (yuan) using the average exchange rate of the previous year, and foreign direct investment was treated as a logarithm. The definitions of the variables and the data sources involved in this study are given in Table 2.

## 5. Results of Quantitative Analysis

### 5.1. Analysis of Baseline Regression Results

This study used a panel fixed-effects model to test the impact of FDI on green economic growth. The regression results are shown in Table 3. Regression (1) only considered the effect of the explanatory variable (FDI) on green economic growth, but the estimation results lacked accuracy, and the level of green economic growth could not be influenced by FDI alone. Regression (3) introduced the core explanatory variable of FDI and all of the related control variables into the model regression analysis. The regression coefficient of FDI was 0.027, which passed the significance test at a 5% level. This indicated that on the whole, foreign capital inflow promoted China’s green economic growth, and Hypothesis H1 was verified. The “pollution halo” hypothesis was validated by the fact that FDI in China could promote green economic growth in China.

In addition, when observing the influence of the control variables on the GTFP, the regression results for the economic development level (PGDP), market size (SCALE), and openness (OPEN) were all significant at a level of 1%. The level of regional economic development positively affected green economic growth. The expansion of market size may have led some enterprises to concentrate on a large amount of production, blindly expand the scale of production, and not carry out technological innovation in time, thereby resulting in serious pollution of the domestic environment that was inconducive to green economic growth. We found that the greater the degree of opening up, the lower the level of green total factor productivity, mainly because the structural differences in import and export commodities had different effects on green economic growth. Therefore, it is very important to optimize China’s import and export structure, reduce resource-intensive and pollution-intensive exports, and increase technology-intensive product trade.

### 5.2. Test of the Mediating Effect of Technological Innovation

The regression results of the mediating effects model are shown in Table 4. The regression results of the intermediary model for autonomous innovation (INNO) showed that firstly, the introduction of foreign capital significantly increased the regional green economic growth (Regression (3): β = 0.027 with *p*-value < 0.05), which satisfied the first condition of the intermediary effect; and secondly, the introduction of foreign capital significantly affected the level of autonomous innovation (Regression (4): β = 0.137 with *p*-value < 0.01), which satisfied the second condition of the test. Subsequently, the level of regional innovation significantly enhanced the regional green economic growth (Regression (5): β = 0.139 with *p*-value < 0.01), which satisfied the third condition of the mediating effect, and the regression coefficient of FDI was no longer significant. Therefore, we suggest that autonomous innovation plays a major mediating role between FDI and China’s green economic growth.

In the intermediation model for imitation innovation (IMMI), firstly, the introduction of foreign capital significantly increased the regional green economic growth (Regression (3): β = 0.027 with *p*-value < 0.05), which satisfied the first condition of the intermediation effect; and secondly, the introduction of foreign capital significantly affected the level of imitation innovation (Regression (6): β = 0.507 with *p*-value < 0.01), which satisfied the second condition of the test. Subsequently, the level of regional imitation innovation significantly inhibited the regional green economic growth (Regression (7): β = −0.01 with *p*-value < 0.05), which satisfied the third condition of the mediating effect, and the regression coefficient of FDI remains significant. This indicated that imitative innovation plays a partially mediating role between FDI and China’s green economic growth.

Based on the overall results of the intermediary effect test, FDI had a positive effect on green economic growth in China, and this promotion could be achieved through the two intermediary variables of independent innovation and imitation innovation (assuming that H2 was established). FDI positively affected green economic growth by promoting autonomous innovation and further inhibited green economic growth by promoting imitative innovation.

### 5.3. Test of the Moderating Effect of Environmental Regulation

#### 5.3.1. The Moderating Effect of Environmental Regulation on FDI Affecting Green Economic Growth

Before the model regression, to avoid the serious multicollinearity problem between the cross-term and other variables, the moderator variable and explanatory variables were centralized and then regressed in this study. The regression results are shown in Table 5 below.

The regression results showed that the regression coefficient of environmental regulation (ER) was negative but insignificant, and there were two possible reasons for the failure of environmental regulation to directly affect regional green economic growth.

First, the current environmental regulation in China is inefficient, lacks sustainability and stability, has the problem of detachment between institutional policy and practice in implementation, and is not able to effectively constrain business organizations.

Second, environmental regulation has a compliance cost effect and an innovation compensation effect, and only when the economic benefits of the latter are greater than the compliance costs of the former can environmental regulation have a positive impact on productivity and green production technologies. The current environmental regulatory policies in China are mainly aimed at achieving pollution control and emission constraints by focusing on energy-intensive and pollution-intensive industries, but lack incentives for high-tech enterprises and organizations to develop green technologies and reform green production processes.

The regression coefficient of the interaction coefficient between environmental regulation (ER) and FDI was positive and significant at a 5% level, which showed that although environmental regulation did not directly affect the level of green economic growth, it could moderate the impact of FDI on green economic growth, which indicated that China as a whole is gradually moving away from the concept of lowering the threshold to attract foreign investment with the establishment of environmental protection awareness. Environmental regulation has steadily improved the quality of FDI inflows to China and brought in new and advanced green technologies and business concepts with the inflow of high-quality foreign investment, thus promoting the positive effects of technology transfer, technology spillover, capital spillover, and environmental spillover of FDI and achieving the coordinated development of economic benefits and ecological environmental protection.

Based on the data in Table 5, environmental regulation had a moderating effect on the relationship between FDI and green economic growth: the greater the environmental regulation, the greater the impact of FDI on green economic growth, so Hypothesis H3 held.

#### 5.3.2. Mediated Moderating Effect Analysis

To further test whether environmental regulation (ER) had a moderating effect on FDI and green economic growth through the mediating effect of technological innovation, we further built a mediating moderating effect model to empirically test Hypotheses H4 and H5. The regression results are shown in Table 6. Before the empirical test, to avoid the multicollinearity problem brought about by the interaction term, we centralized the moderator variable and explanatory variables before regression. This study drew on the test methods of Wen and other scholars [32].

The regression results of the mediation model of independent innovation (INNO) showed that firstly, the interaction term between foreign direct investment and environmental regulation significantly improved the regional green economic growth (Regression (10): regression coefficient 0.002, *p* < 0.05), which satisfied the first step of the test, thereby indicating that environmental regulation positively regulated the role of FDI in promoting green economic growth. Secondly, according to Regression (11), the interaction between foreign direct investment and environmental regulation had no significant impact on the level of regional independent innovation; that is, environmental regulation could not affect the impact of FDI on independent innovation, so it can be considered that environmental regulation did not play a regulatory role through the intermediary effect of independent innovation.

The regression results of the mediated moderation model of imitation innovation (IMMI) showed that firstly, the interaction between foreign direct investment and environmental regulation significantly improved the regional green economic growth (Regression (10): regression coefficient 0.002, *p* < 0.05), which satisfied the first step of the test. Secondly, the interaction between foreign direct investment and environmental regulation significantly affected the level of regional imitation innovation (Regression (12): regression coefficient 0.035, *p* < 0.1), which satisfied the second condition of the test; that is, environmental regulation could positively regulate the relationship between FDI and imitation innovation. Subsequently, the level of regional imitation innovation significantly inhibited the growth in the regional green economy (Regression (13): regression coefficient-0.01, *p* < 0.01), which satisfied the third condition of the test, thereby indicating that the regulatory role of environmental regulation on FDI and green economic growth could be played partly through imitation innovation.

According to the above analysis, it can be seen that environmental regulation could positively regulate the impact of FDI on imitative innovation, thus affecting China’s green economic growth, and that there was no significant moderating effect on the intermediary transmission path of independent innovation. H4 was established, but H5 was not established.

### 5.4. Robustness Tests

To avoid the omission of variables and because there may have been some endogeneity problems between FDI, the different technological innovation models, and green economic growth, this study took FDI lagged by one period as the instrumental variable and used the panel instrumental variable method to regress the above research again to ensure the robustness of the results.

As can be seen in Table 7, the results of the robustness tests were generally consistent with the above except for the differences in the coefficients, and the sign and significance of the coefficients of each variable remained unchanged, thereby indicating that the estimation results were robust.

## 6. Discussion

This study revisited the green growth effect of foreign direct investment from an environmental regulation perspective. The research contributions were as follows.

Firstly, this study verified that FDI can significantly contribute to green economic growth in China, thereby proving the “pollution halo” hypothesis. This was consistent with the findings of several scholars. Among them, Birdsall [33] studied the relationship between the liberalization of trade regimes, the increase in foreign investment, and the development of pollution-intensive industries in Latin America and concluded that openness can encourage clean industries by introducing the pollution standards of developed countries. Yue [34] used a sample of 104 cities in China to demonstrate that FDI promoted green growth in Chinese cities through environmental and economic benefits. Zhou and Zhang [35] built a logical framework of “FDI-economic agglomeration-green economic efficiency” and showed that FDI promoted local green economic efficiency through economic agglomeration and energy-saving and emission-reduction technologies. However, the findings of this study were contrary to those of Xia [36], who argued that foreign investors have transferred a large number of pollution-intensive industries to China through FDI, which has seriously inhibited the process of green transformation of China’s economy. This may be because local governments have become more stringent in approving the introduction of non-clean FDI in recent years. The inflow of clean FDI has to some extent promoted the optimization and upgrading of regional industrial structure and improved the efficiency of resource allocation and use, thus contributing to the improvement in the GTFP.

In addition, this paper constructed a research framework of “FDI-technology innovation-GTFP”. Among the existing studies, Liu and Li [37,38] demonstrated the positive impact of FDI on technological innovation in host countries. Meanwhile, Chen, Fu, and Wang [39,40,41] argued that technological innovation can increase green total factor productivity. Based on the existing literature, this study constructed and proves the mechanism of the impact of FDI on GTFP.

Due to data limitations and other reasons, this study still had some limitations. First, when measuring the green total factor productivity using MAXDEA software, the input indicators, expected output indicators, and non-expected output indicators used were not comprehensive enough, so the measurement of green economic growth level still needs to be further optimized. Second, in an actual situation, the impact of FDI with different characteristics on green economic growth has variability. Due to the limitation of the article length, FDI was not divided in this paper. Third, this study explored the relationship between FDI, technological innovation, green economic growth, and environmental regulation based on macro data at the provincial level, so the macro-level findings may not fully satisfy the needs of micro decision-making behavior.

## 7. Conclusions

In order to promote green total factor productivity and achieve sustainable economic growth in China, this study investigated the green economic growth effect of FDI from the perspective of environmental regulation. Based on the interprovincial panel data for China from 2004 to 2019, we measured the level of green economic growth in China using the SBM-GML index and empirically analyzed the impact of FDI on green economic growth in China and its mechanism of action. Our main conclusions were as follows: First, FDI significantly promoted China’s green economic growth. Second, there was an FDI-autonomous innovation (imitation innovation) to green economic growth path; FDI inhibited green economic growth via imitation innovation and promoted green economic growth via autonomous innovation; third, environmental regulation affected the impact of FDI on regional green economic growth, and the regulating effect was partly mediated by the transmission mechanism of imitation innovation. In summary, this study constructed and demonstrates the transmission mechanism of “foreign direct investment-technology innovation-green total factor productivity” and further verified the moderating role of environmental regulation in this mechanism, which provided a theoretical basis and practical reference for achieving green economic growth in China.

We established the following policy recommendations based on the above findings. First, China is currently in an important stage of economic transformation, and economic growth emphasizes both scale and quality. To avoid foreign companies from regarding China as a “pollution refuge” for their production and operation, local governments should make some trade-offs in the introduction of foreign capital. They should guide the inflow of high-quality foreign investment, control the entry of FDI into China’s resource-intensive and pollution-intensive industries, and raise the entry standards for foreign investment. Second, to strengthen independent innovation and reduce technological dependence, government departments should take corresponding measures to actively guide enterprises to establish a sense of crisis, carry out independent innovation, reduce imitation innovation, and maintain a sense of urgency and responsibility for the world’s frontier technological breakthroughs. Third, China should enrich the means of environmental regulation to achieve effective restraint. Reasonable environmental regulation can help bring into play the positive effect of FDI on the development of China’s green economy. Local governments should strictly implement environmental regulation policies and formulate diversified control forms in combination with aspects of the actual situation.

## Figures and Tables

**Figure 1 ijerph-20-02655-f001:**
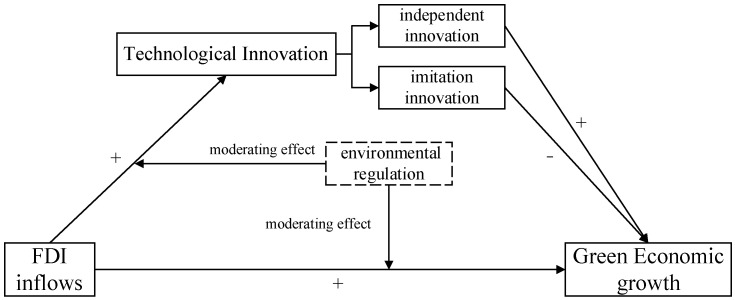
The mechanism of FDI’s influence on green economic growth.

**Table 1 ijerph-20-02655-t001:** The measurement indicators of GTFP.

	Variable	Definition
Input	Labor input	Number of employees at the end of the year in the region.
	Capital input	The capital stock as estimated via the perpetual inventory method.
	Energy input	Total energy consumption according to province.
Output	Desirable output	GDP of each province.
	Undesirable output	General industrial solid-waste production Industrial wastewater emissions Carbon dioxide emissions Industrial sulfur dioxide emissions

**Table 2 ijerph-20-02655-t002:** The list of variables and data sources.

Variable	Definition	Source(s)	Mean	S.D.	Min	Max
Explained variable						
GTFP	This study assumed that the green total factor productivity (GTFP) in 2004 was 1 and then calculated the green total factor productivity from 2004 to 2019 by multiplying the GML index by the GTFP of the last year.	*China Energy Statistical Yearbook*, *China Statistical Yearbook on Environment*, and CEADS database	1.3850	0.3270	1.0130	3.6280
Explanatory variable						
FDI	Foreign direct investment (FDI) was measured according to the proportion of actual utilization of foreign investment in the GDP.	Wind database	0.0250	0.0220	0.0001	0.1210
Moderating variable						
ER	Environmental regulation (ER) was calculated according to the comprehensive utilization rate of industrial solid waste, the ratio of the operation cost of industrial waste gas treatment facilities to the industrial waste gas emissions, and the ratio of the operation cost of industrial wastewater treatment facilities to the industrial wastewater emissions.	*China Statistical Yearbook on Environment*	8.9540	5.2770	1.2460	54.7930
Mediator variables						
INNO	Independent innovation (INNO) was measured according to the number of invention patent authorizations.	*China Statistical Yearbook on Science and Technology*	0.4920	0.8870	0.0020	5.9740
IMMI	Imitation innovation (IMMI) was measured according to the sum of appearance patent authorizations and practical patent authorizations.	*China Statistical Yearbook on Science and Technology*	2.9250	5.4380	0.0050	46.7650
Control variables						
SCALE	Market size (SCALE) was measured according to the total population of each region.	National Bureau of Statistics	0.4480	0.2740	0.0540	1.2490
INFRA	Infrastructure security (INFRA) was measured according to the ratio of the total post and telecommunications business to the GDP.	National Bureau of Statistics	0.0630	0.0340	0.0200	0.2360
GOVER	Government support (GOVER) was measured according to the proportion of regional fiscal expenditures on science and technology.	National Bureau of Statistics	0.0180	0.0140	0.0010	0.0720
OPEN	The degree of openness (OPEN) was measured according to the ratio of the total imports and exports of business units to the GDP in each region.	National Bureau of Statistics	0.3110	0.3470	0.0110	1.6640
EDU	The level of education (EDU) was measured according to the average number of years of education = (the number of people with literacy skills × 1 + the number of people with elementary school education × 6 + the number of people with junior high school education × 9 + the number of people with high school and junior college education × 12 + the number of people with college and undergraduate education × 16)/total population over 6 years old.	National Bureau of Statistics	8.7650	1.0170	6.3780	12.7820
SOE	The degree of nationalization (SOE) was measured according to the number of state-owned industrial enterprises divided by the number of industrial enterprises.	National Bureau of Statistics	0.1240	0.0920	0.0110	0.4330
PGDP	The level of economic development (PGDP) was measured according to the per capita GDP.	National Bureau of Statistics	0.4480	0.2740	0.0540	1.2490

**Table 3 ijerph-20-02655-t003:** Baseline regression results.

	(1)	(2)	(3)
	GTFP	GTFP	GTFP
FDI	0.011 (0.014)		0.027 ** (0.013)
PGDP		0.144 *** (0.021)	0.145 *** (0.021)
SCALE		−1.272 *** (0.340)	−1.294 *** (0.346)
OPEN		−0.226 *** (0.086)	−0.245 *** (0.087)
SOE		0.224 (0.205)	0.230 (0.204)
EDU		0.057 (0.043)	0.059 (0.043)
GOVER		0.162 (1.650)	−0.306 (1.732)
INFRA		−0.121 (0.646)	0.004 (0.651)
Constant	1.430 *** (0.059)	0.941 ** (0.443)	1.052 ** (0.432)
Province fixed effect	Yes	Yes	Yes
Time fixed effect	Yes	Yes	Yes
N	480	480	480
R^2^	0.749	0.870	0.871

Note: ** *p* < 0.05, *** *p* < 0.01.

**Table 4 ijerph-20-02655-t004:** The mediating effect of independent innovation and imitation innovation.

	(4)	(5)	(6)	(7)
	INNO	GTFP	IMMI	GTFP
INNO		0.139 *** (0.036)		−0.010 ** (0.004)
FDI	0.137 *** (0.028)	0.008 (0.011)	0.507 *** (0.180)	0.032 ** (0.014)
PGDP	0.332 *** (0.044)	0.099 *** (0.018)	0.348 (0.260)	0.149 *** (0.021)
SCALE	9.496 *** (1.141)	−2.616 *** (0.558)	83.019 *** (11.717)	−0.485 (0.468)
OPEN	−0.211 (0.234)	−0.215 ** (0.097)	−2.606 * (1.527)	−0.270 *** (0.081)
SOE	0.302 (0.517)	0.188 (0.183)	4.499 (3.519)	0.274 (0.206)
EDU	0.156 * (0.092)	0.037 (0.038)	0.881 (0.577)	0.067 (0.043)
GOVER	4.053 (3.439)	−0.870 (1.715)	45.848 * (24.886)	0.141 (1.668)
INFRA	2.059 (1.590)	−0.283 (0.532)	5.486 (8.301)	0.057 (0.666)
Constant	−6.032 *** (0.985)	1.892 *** (0.430)	−42.211 *** (7.385)	0.641 (0.515)
Province FE	Yes	Yes	Yes	Yes
Time FE	Yes	Yes	Yes	Yes
N	480	480	480	480
R^2^	0.907	0.884	0.892	0.874

Note: * *p* < 0.10, ** *p* < 0.05, *** *p* < 0.01.

**Table 5 ijerph-20-02655-t005:** Moderating effect of environmental regulation on FDI.

	(8)	(9)
	GTFP	GTFP
FDI	0.028 ** (0.013)	0.032 ** (0.014)
ER	−0.001 (0.001)	−0.001 (0.001)
FDI × ER		0.002 ** (0.001)
PGDP	0.145 *** (0.021)	0.143 *** (0.021)
SCALE	−1.291 *** (0.345)	−1.238 *** (0.344)
OPEN	−0.246 *** (0.087)	−0.219 *** (0.084)
SOE	0.240 (0.206)	0.260 (0.207)
EDU	0.056 (0.043)	0.056 (0.043)
GOVER	−0.312 (1.731)	−0.575 (1.784)
INFRA	−0.015 (0.656)	−0.091 (0.656)
Constant	0.958 ** (0.438)	0.942 ** (0.436)
Province FE	Yes	Yes
Time FE	Yes	Yes
N	480	480
R^2^	0.871	0.873

Note: ** *p* < 0.05, *** *p* < 0.01.

**Table 6 ijerph-20-02655-t006:** The mediated moderating effect of environmental regulation.

	(10)	(11)	(12)	(13)
	GTFP	INNO	IMMI	GTFP
IMMI				−0.010 *** (0.004)
FDI	0.032 ** (0.014)	0.133 *** (0.029)	0.574 *** (0.193)	0.038 *** (0.014)
ER	−0.001 (0.001)	−0.003 (0.003)	0.016 (0.023)	−0.001 (0.001)
FDI × ER	0.002 ** (0.001)	−0.002 (0.003)	0.035 * (0.019)	0.003 ** (0.001)
PGDP	0.143 *** (0.021)	0.332 *** (0.044)	0.329 (0.263)	0.147 *** (0.021)
SCALE	−1.238 *** (0.344)	9.463 *** (1.138)	83.713 *** (11.809)	−0.371 (0.450)
OPEN	−0.219 *** (0.084)	−0.237 (0.226)	−2.216 (1.558)	−0.242 *** (0.079)
SOE	0.260 (0.207)	0.320 (0.519)	4.624 (3.506)	0.308 (0.210)
EDU	0.056 (0.043)	0.148 (0.093)	0.909 (0.572)	0.065 (0.043)
GOVER	−0.575 (1.784)	4.248 (3.465)	42.272 * (24.557)	−0.137 (1.711)
INFRA	−0.091 (0.656)	2.060 (1.620)	4.685(8.359)	−0.043 (0.669)
Constant	0.942 ** (0.436)	−6.515 *** (1.006)	−44.810 *** (7.470)	0.478 (0.521)
Province fixed effect	Yes	Yes	Yes	Yes
Time fixed effect	Yes	Yes	Yes	Yes
N	480	480	480	480
R^2^	0.873	0.908	0.893	0.876

Note: * *p* < 0.10, ** *p* < 0.05, *** *p* < 0.01.

**Table 7 ijerph-20-02655-t007:** Robustness tests.

	GTFP	GTFP	GTFP	INNO	GTFP	IMMI	GTFP
FDI	0.042 **	0.043 **	0.049 ***	0.159 ***	0.019	0.516 *	0.047 ***
	(0.018)	(0.018)	(0.018)	(0.040)	(0.015)	(0.284)	(0.018)
INNO					0.145 ***		
					(0.035)		
IMMI							−0.009 **
							(0.004)
ER		−0.001	−0.001				
		(0.001)	(0.001)				
FDI × ER			0.003 **				
			(0.001)				
PGDP	0.144 ***	0.144 ***	0.142 ***	0.332 ***	0.096 ***	0.274	0.147 ***
	(0.020)	(0.020)	(0.020)	(0.042)	(0.017)	(0.255)	(0.020)
SCALE	−1.409 ***	−1.407 ***	−1.332 ***	9.928 ***	−2.848 ***	87.347 ***	−0.656
	(0.368)	(0.366)	(0.366)	(1.125)	(0.572)	(12.004)	(0.504)
OPEN	−0.250 ***	−0.252 ***	−0.221 ***	−0.250	−0.214 **	−2.958 *	−0.276 ***
	(0.088)	(0.089)	(0.085)	(0.244)	(0.101)	(1.576)	(0.081)
SOE	0.639 ***	0.657 ***	0.711 ***	1.540 **	0.416 **	7.439 **	0.703 ***
	(0.235)	(0.239)	(0.237)	(0.604)	(0.206)	(3.734)	(0.242)
EDU	0.063	0.060	0.061	0.217**	0.031	1.245 **	0.074 *
	(0.041)	(0.041)	(0.041)	(0.095)	(0.036)	(0.531)	(0.043)
GOVER	−1.095	−1.108	−1.431	4.943	−1.811	61.537 **	−0.564
	(1.930)	(1.925)	(1.998)	(3.573)	(1.926)	(28.391)	(1.900)
INFRA	−0.073	−0.097	−0.199	1.675	−0.316	2.319	−0.053
	(0.613)	(0.616)	(0.611)	(1.482)	(0.501)	(7.858)	(0.623)
N	450	450	450	450	450	450	450
R^2^	0.677	0.677	0.681	0.831	0.713	0.765	0.682

Note: * *p* < 0.10, ** *p* < 0.05, *** *p* < 0.01.

## Data Availability

The data presented in this research are available on request from the corresponding author.

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
