# Peer review of "Revisiting the Green Growth Effect of Foreign Direct Investment from the Perspective of Environmental Regulation: Evidence from China"

_ijerph, 2023, doi:10.3390/ijerph20032655_

Round 1
Reviewer 1 Report
The role of FDI in green economic growth is an important topic, worthy of thorough investigation. This is why I think the authors could have given an opportunity to significantly improve the quality of the manuscript. I would bring out the following aspects that need to be addressed:
- The manuscript needs editing by native speaker of English. There are some odd expressions that repeatedly occur, e.g. 'the literature ...' instead of 'the paper ...' (see line 11 and elsewhere).
- In places, the authors use first sentence of a paragraph as a subtitle (e.g. lines 54, 62 and elsewhere). I don't think this is a good style. If authors want to highlight these subtitles, these should be formatted as subtitles.
- Introduction is currently weak. It would benefit if the authors merge the contents of introduction and literature review, and thoroughly rework the presentation of the issues. More background information is needed on the role of FDI in the current economic growth model. Also, the concept of green economic growth and pollution paradise hypothesis needs to explained more thoroughly. Research gap and the objective of the paper needs to be presented more clearly.
- In text (lines 91-97) the authors refer to 'many scholars' without providing any references. In such cases, clearly, relevant references should be provided. Lack of references is problem also in lines 112-119.
- Hypothesis H2 incorporates too many different relationships, and therefore cannot considered as a hypothesis. '... different technological innovation models have different mediating 207 effects on FDI on green economic growth.' cannot be a hypothesis because it remains the door open for any kind of relationships between infinite number of variables.
- Similarly, hypotheses H4 and H5 needs to be refined. Achieving something partly is not specific enough. See lines 234 and 239: 'The moderating effect of environmental regulation on the relationship between FDI and green economic growth is partly achieved through the mediating mechanism of imitation innovation.'
- It seems that not all the effects depicted in Figure 1 are explained in the text, thereby raising the question to which extent Figure 1 actually illustarates the mechanisms that are studied in this mansucript. The authors should improve the text and/or Figure 1 respectively.
- Equation (2) does not clearly show how 'The TECHit represents the technological innovation in region i at time t, which is subdivided into independent innovation (INNOit) and imitation innovation (IMMIit).' (lines 265-266).
- Since a separate model is used to measure GTFP, more information on the model and its estimation should be provided.
- Please reword explanation of the explanatory variable FDI (lines 304-305). It is not clear how this is measured.
- Also, please provide more details hoe the ER variable is calculated (lines 307-313).
The following variables need to be explained and defined more clearly: 'ratio of total post and telecommunications business to GDP' (lines 323-324); proportion of regional fiscal expenditures on science and technology (lines 325-326); three costs in 'Due to the change of the statistical caliber of science and technology expenditures,the data from 2004 to 2006 are the three costs of science and technology, and the other years are the costs of science and technology expenditures' (lines 326-328).
- I would suggest to add an annex with tescriptive statistics of all the variables used in the study.
- The manuscript lacks any discussion of the achieved results in comparison with other studies of green economic growth. This shortcoming should be considered. Please explain how your results fit in the overall international discourse on the topic.
- Presentation of results are confusing. It seems that estimation results of model (2) are not presented, while estimations of models (3) in Table 1, model (4) in Table 2, and model (7) in Table 3 are exactly the same. So, I would say that the numbering of estimation results does not correspond to the numbering of models presented in methodology chapter. This should be revised and corrected.
- In Table 5, there are no estimation results for the INNO variable. Therefore, the INNO variable should be removed from this Table.
Reviewer 3 Report
Abstract
1. The research background and limitations of previous studies are described in 1-2 sentences.
2. The research methods of the manuscript should be added to the abstract section.
3. The conclusion of the abstract should increase the revelation and significance of the manuscript.
Introduction
4. Research methods, innovations and theoretical basis should be added to the introduction part of the manuscript, and the author should make a supplementary explanation.
5. Since the acronym “FDI” in this part appears for the first time in the manuscript, its full name should be expressed and corresponding explanation given.
6. Authors are advised to ask scientific questions that the manuscript needs to address in the form of questions.
7. Although the authors list the views of several researchers, a broad and in-depth review of the critical literature is lacking. In addition, some important literature in the field of technological innovation has been omitted.
e.g.,
Chen, H., Yi, J., Chen, A., Peng, D., & Yang, J. (2023). Green technology innovation and CO2 emission in China: Evidence from a spatial-temporal analysis and a nonlinear spatial durbin model. Energy Policy, 172, 113338.
Fu, K., Li, Y., Mao, H., & Miao, Z. (2023). Firms’ production and green technology strategies: The role of emission asymmetry and carbon taxes. European Journal of Operational Research, 305(3), 1100-1112.
Li, X., Huang, Y., Li, J., Liu, X., He, J., & Dai, J. (2022). The mechanism of influencing green technology innovation behavior: evidence from Chinese construction enterprises. Buildings, 12(2), 237.
Wang, S., Li, J., & Razzaq, A. (2023). Do environmental governance, technology innovation and institutions lead to lower resource footprints: An imperative trajectory for sustainability. Resources Policy, 80, 103142.
etc.
Research methods
8. Lack of explanation of the merits, applicability and reasons for not choosing alternative methods.
9. It is suggested that the control variables should be listed in a table, and the control variables should be explained in detail.
10. Please add a corresponding comment for the data source section.
Disscusions
​11. It is suggested that the author add a section after the result analysis to discuss whether other similar studies are consistent with the results of the manuscript research. If not, the reasons should be analyzed and the limitations and future prospects of the manuscript should be expressed.
Conclusion and suggestions
12. Please supplement the research purpose of the manuscript in the conclusion. At the same time, please make a summary of each point summary, come up with a general summary, to describe the overall situation.
13. In addition, don't use ‘——' in the title, it's not English. Perhaps you should easily proofread this manuscript for native English speakers.
To sum up, it is suggested that the authors carefully revise this manuscript according to the above suggestions. I sincerely look forward to receiving the revised version.
Round 2
Reviewer 1 Report
Thanl you for the revised and much improved version of the manuscript. There are still some issues that could be improved before publication:
- Please explain the concept of green economic growth also in the introduction. This is very much related to the aims of this paper and not properly introduced before stating the objectives.
- In the theoretical analysis chapter, on lines 217-254, it is unclear whether the provided information is 100% created by the authors or there is information that is available also in the other literature sources. I would encourage authors to add references to previous literature on the introduced concepts (e.g. capital effect of FDI, technology spillover effect of FDI, technology transfer).
- I would recommend to limit the wording of hypotheses H2 and H4 only with the specific part and remove general statements.
- Please explain also model (3).
- From the model (2) it is not clear that INNOit and IMMIit are verisions of Techit. Another subscript should be added to Techit that refers to INNOit and IMMIit.
- In Table 1 it is unclear if undesirable outpout variable is one variable or four variables, as it is defined through four variables.
- As in the previous review, I would encourage authors to add descriptive statistics (min, max, mean, standard deviation) of the variables listed in Table 2.
- In Table 2, it is not clear what variables were multiplied to obtain GTFP, this should be revised.
- The title of chapter 5 should be revised (e.g. Results of quantitative analysis).
Reviewer 3 Report
The authors have modified the paper as suggested, so it could be accepted in present form.
